# TimelinePTC: Development of a unified interface for pathways to care collection, visualization, and collaboration in first episode psychosis

**Walter S. Mathis**[1,2]☯*, **Maria Ferrara**[2,3], **John Cahill**[1,2], **Sneha Karmani**[1,2‡], **Sümeyra N. Tayfur**[1,2‡], **Vinod Srihari**[1]☯

**1** Department of Psychiatry, Yale University School of Medicine, New Haven, Connecticut, United States of America, **2** Program for Specialized Treatment Early in Psychosis (STEP), New Haven, CT, United States of America, **3** Department of Neuroscience and Rehabilitation, Institute of Psychiatry, University of Ferrara, Ferrara, Italy

☯ These authors contributed equally to this work.
‡ These authors also contributed equally to this work
* walter.mathis@yale.edu

**Data Availability Statement:** https://github.com/StanMathisYale/TimelinePTC

**Funding:** None of the authors have any conflicts of interest of financial support to report. This work

## Abstract

This paper presents TimelinePTC, a web-based tool developed to improve the collection and analysis of Pathways to Care (PTC) data in first episode psychosis (FEP) research. Accurately measuring the duration of untreated psychosis (DUP) is essential for effective FEP treatment, requiring detailed understanding of the patient's journey to care. However, traditional PTC data collection methods, mainly manual and paper-based, are time-consuming and often fail to capture the full complexity of care pathways. TimelinePTC addresses these limitations by providing a digital platform for collaborative, real-time data entry and visualization, thereby enhancing data accuracy and collection efficiency. Initially created for the Specialized Treatment Early in Psychosis (STEP) program in New Haven, Connecticut, its design allows for straightforward adaptation to other healthcare contexts, facilitated by its open-source codebase. The tool significantly simplifies the data collection process, making it more efficient and user-friendly. It automates the conversion of collected data into a format ready for analysis, reducing manual transcription errors and saving time. By enabling more detailed and consistent data collection, TimelinePTC has the potential to improve healthcare access research, supporting the development of targeted interventions to reduce DUP and improve patient outcomes.

## Introduction

Understanding the duration of untreated psychosis (DUP) is crucial in the treatment of first episode psychosis (FEP) [1–3]. DUP, the time from the onset of psychosis symptoms to the initiation of specialized treatment, has been identified as a critical factor influencing the outcomes of treatment [4–6]. Consequently, there is a pressing need to not only quantify DUP

was supported by National Institutes of Health (R01MH103831) and the Gustavus and Louise Pfeiffer Research Foundation. The funding sources had no role in the design and conduct of the study; collection, management, analysis, and interpretation of the data; preparation, review, or approval of the manuscript; and decision to submit the manuscript for publication. This work was also funded by the State of Connecticut, Department of Mental Health and Addiction Services, but this publication does not express the views of the Department of Mental Health and Addition Services or the State of Connecticut. The views and opinions expressed are those of the authors.

**Competing interests:** No authors have competing interests

but also to dissect and understand the pathways to care (PTC), the saltatory linkage of community and clinical interactions individuals travel from symptom onset to ultimate treatment [7]. Analyzing these pathways provides invaluable insights into potential delays and barriers both outside and within the healthcare system and can inform interventions to improve the experience of seeking care and reducing the delays in accessing it [8].

Despite the recognized importance of mapping pathways to care, significant gaps exist in both data collection and analysis. Surprisingly, most clinics providing specialized care to those experiencing FEP, which are specifically designed to intervene early in the illness course, fail to collect any PTC data [9]. Of the few that do report on PTC, most collect or analyze their data in ways that significantly limit inferential power. These approaches typically involve manually reconstructing pathways either from electronic medical records (EMR) alone or through patient interviews supplemented by EMR review, often without a standardized format [10,11]. This *ad hoc* methodology results in significant variability in the quality and granularity of the data collected, making it difficult to compare across studies or to draw broad conclusions about barriers to care. Further, the PTC data collected for these studies are not scoped to capture the patient's full journey to care, instead focusing on a few key time measures (e.g. help-seeking delay). And these measures themselves are determined *a priori*, not suggested by, or inferred from the data.

Even within the subset of studies that prioritize the collection of high-quality PTC data, the prevailing reliance on paper-based tools poses significant challenges [8,12]. These traditional methods of data collection are often time-consuming and designed with a research-centric perspective, rather than being patient-focused. Moreover, they typically impose a predetermined structure on the data being collected, such as categorizing experiences into "episodes of care" [13]. This approach, while systematic, may not necessarily capture the full complexity or nuances of individual pathways to care, potentially overlooking critical aspects of the patient's journey that are vital for a comprehensive analysis. While there are rare instances of studies attempting to innovate within this space—notable examples being a single study that adopted digital forms for data collection [14] and another unique approach utilizing a paper visual timeline to facilitate collaborative data collection with participants [15]—these remain exceptions rather than the norm. Such efforts to modernize PTC data collection highlight the existing gap and the need for more interactive, flexible, and patient-centered tools that collect granular, complete, and valid PTC data.

Therefore, we introduce TimelinePTC, designed specifically to overcome these challenges. Built to be digital, visual, and interactive, TimelinePTC changes how pathways to care data are collected. It allows for real-time, joint data entry and visualization between researchers and participants, improving both the quality and accuracy of the data while making the process more efficient. TimelinePTC provides a flexible and easy-to-use platform that can be adjusted for different analytic needs, significantly enhancing our ability to understand and address pathways to care in first episode psychosis.

## Software description

TimelinePTC is a data collection and visualization tool, engineered to capture the complex journeys patients undertake from the onset of symptoms to their enrollment in specialty care.

Opening the interface, users are prompted to input critical baseline information, including a participant ID, an estimated date of symptom onset (later confirmed via a separate structured assessment), and the date of enrollment in the FEP clinic (Fig 1A). These data then frame the interactive timeline that stretches from the reported onset of illness to clinic enrollment, with months and years marked as reference points (Fig 1B).

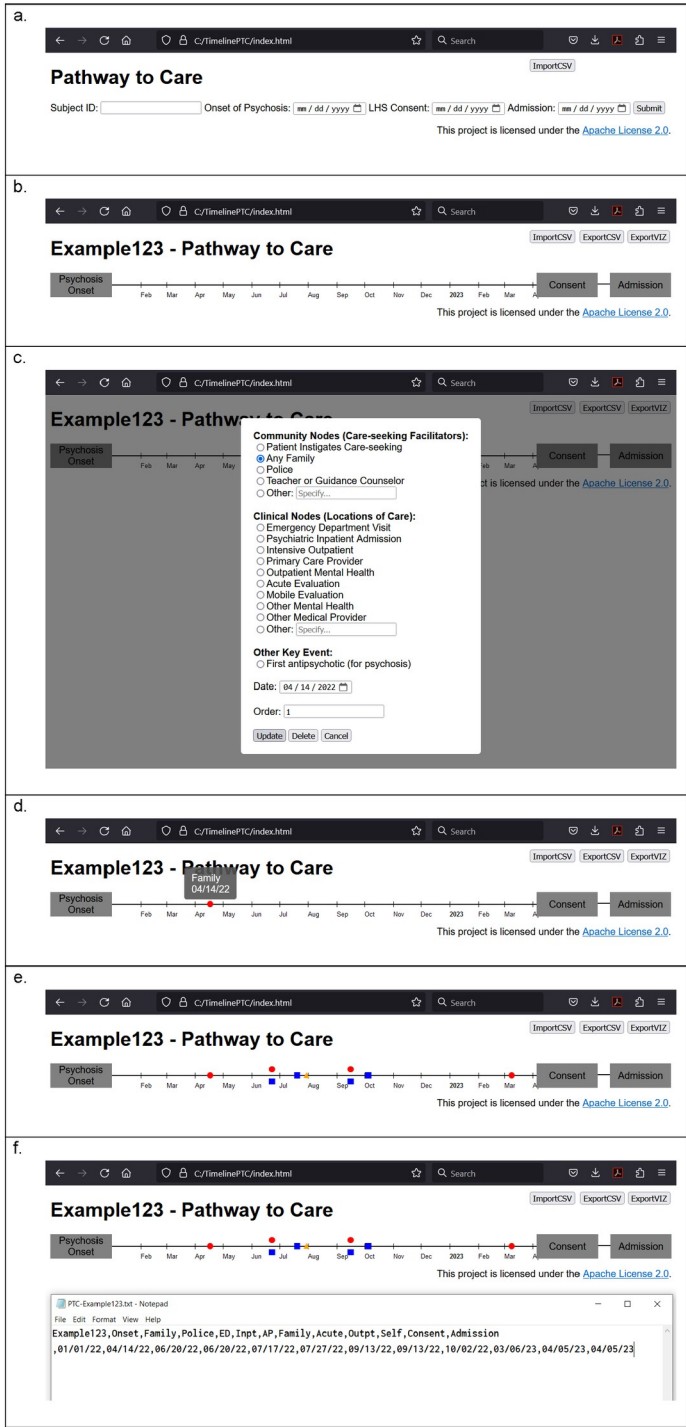

**Fig 1. TimelinePTC in different phases of use.** 1a. Initial screen for baseline information 1b. The timeline drawn from these dates 1c. The interface that pops up when the timeline is clicked 1d. The timeline showing an event that has been entered 1e. A more robust pathway to care 1f. PTC data extracted to comma-separated values (CSV) format.

The user interface is simple, offering an intuitive timeline as the centerpiece for both data entry and visualization. This timeline is not just a passive display; it is a canvas for interaction. The research assistant, using a semi-structured intake script, prompts for key events on the

timeline [See S1 Table]. Language in this script is carefully constructed to collect interactions that are seen to be part of the pathway to care for FEP, that is events that were specifically relevant to recognition and facilitation of care of their FEP symptoms [see S1 Table for details].

These events are added to the timeline by clicking on the corresponding point along the timeline which opens a popup window where more detailed information about the event can be entered (Fig 1C). The types of events are adapted from those empirically derived from our previous PTC analyses [8] and are grouped into clinical nodes (individuals or agencies providing clinical care–e.g. emergency departments, primary care providers, therapists) and community nodes (those with the capacity to facilitate access to treatment–e.g. family members, police officers, teachers). These node labels are provincial, in that they reflect those most utilized in PTCs collected from STEP, and not necessarily universal. Hence, there is also an option for "other" node types not explicitly listed as well as the opportunity to edit the freely available project source code to tailor to other care settings.

First antipsychotic prescribed for psychosis is its own event class as it serves to demarcate the transition between care epochs in subsequent analysis. A patient's history with other psychotropic medications is certainly important to clinical care. But it has not been included in our PTC analysis nor those by other research teams, and was not included as data collected in TimelinePTC.

As the clinic staff uses the intake script to probe for information, responses are visualized in real-time on the timeline through the addition of icons representing different events (Fig 1D and 1E). This iterative process not only ensures comprehensive coverage of the patient's healthcare journey but also engages the participant in the process, making it more collaborative and transparent. This dynamic interaction facilitated by TimelinePTC encourages a deep dive into the patient's experiences, prompting recollection and discussion of both antecedent and subsequent healthcare interactions. The live visual representation of the data not only aids in ensuring accuracy and completeness but also provides the patient and clinic staff with a holistic view of the healthcare journey as it unfolds.

In addition to its robust in-person data collection capabilities, TimelinePTC seamlessly adapts to the evolving landscape of healthcare by integrating smoothly with telehealth settings. Utilizing screen-sharing features of commonly used telehealth platforms, TimelinePTC allows for real-time, remote collaboration between clinic staff and participants. This adaptation not only maintains the interactive and engaging nature of the tool but also extends its accessibility, ensuring that data collection and visualization can proceed uninterrupted, regardless of physical distance. This integration with telehealth technologies underscores TimelinePTC's flexibility and commitment to facilitating comprehensive care pathway analysis in a variety of clinical environments, further enhancing its utility in modern healthcare contexts.

Once the data collection phase is complete, one of TimelinePTC's most powerful features comes into play: the ability to instantly export the collected data into a digital format optimized for downstream analytics (Fig 1F). This seamless transition from data collection to analysis represents a key advantage, eliminating the time-consuming and error-prone process of translating paper-based records into digital data.

## Development and implementation

### Development

TimelinePTC was developed as part of an ongoing research study on first episode psychosis at the Specialized Treatment Early in Psychosis (STEP) program in New Haven, Connecticut that had been manually collecting PTC data for nearly a decade. The development was driven by two main needs: the existing PTC scale did not capture all the necessary data for robust,

granular delay analysis, and the manual transcription of paper data into a digital format was onerous.

A preliminary version of the data collection and visualization interface was created by a team member (WSM) with a focus on data analysis. This version was then refined through workshops with the research team, including those experienced in data collection from study participants, who provided practical feedback on its functionality. Following integration of this feedback, TimelinePTC was pilot tested with participants newly entering the study, which helped identify technical issues and gather further suggestions for improvement. This pilot testing occurred between January 26 and March 19, 2024. All subjects provided written, informed consent within a protocol approved by the Yale Human Investigations Committee.

Alongside the development of the software, a semi-structured interview format was also created to better guide the data collection process [See S1 Table]. After testing and refinement, TimelinePTC was adopted to replace the traditional paper-based PTC data collection method in the study.

## Technical framework

TimelinePTC is coded in JavaScript and HTML, technologies chosen for their universal compatibility and ease of deployment across a wide range of devices. This technical framework ensures that TimelinePTC can be effortlessly accessed and utilized on various platforms, from smartphones and tablets to laptops and desktop computers, without the need for any specialized software–merely a web browser is sufficient. This also supports flexibility for data collection in diverse environments, including clinics, research laboratories, and fieldwork settings.

All computer code is run locally, ensuring that sensitive data, such as protected health information including dates of healthcare services rendered, never leave the local device. This allows clinics and research facilities to decide the most appropriate method for managing the data output–e.g. saving on a secure, local computer system, transferring to an encrypted thumb drive, or opting for storage in a HIPAA-compliant cloud-based repository.

## Design philosophy

The design philosophy behind TimelinePTC is rooted in the principles of speed, portability, and clarity, aimed at facilitating a seamless and efficient user experience. The software's visual language is intentionally sparse, drawing inspiration from PTC visualizations developed in our prior analyses [8]. This minimalist approach is not merely an aesthetic choice but a deliberate effort to put the emphasis squarely on the data being collected and displayed. By reducing visual clutter, TimelinePTC ensures that clinic staff and participants can easily interpret and evaluate the information without distraction, fostering a clear and focused environment for data entry and review.

Moreover, the straightforward and unobtrusive interface of TimelinePTC is crafted to keep users engaged in their cooperative task without becoming preoccupied with navigating or manipulating the software itself. The aim is to create a digital space where technology serves as a silent partner in the data collection process, enhancing the interaction between research assistants and participants rather than overshadowing it. This approach ensures that the software's functionality enhances the collaborative experience, making TimelinePTC not just a tool for data collection, but a facilitator of meaningful dialogue and discovery in the pathway to care research.

## Accessibility and licensing

TimelinePTC has been developed with a strong commitment to accessibility and open collaboration. It is currently hosted on a public GitHub repository, readily accessible to all who wish to utilize this innovative tool for pathways to care data collection and visualization (https://github.com/StanMathisYale/TimelinePTC). This not only facilitates ease of access but also encourages contributions from the global development community, allowing for enhancements, customizations, and potentially new features driven by user feedback and collaborative innovation.

To align with our goals of open use and contribution while ensuring the original development team is credited appropriately, TimelinePTC is released under the Apache License 2.0. This license provides the freedom to use, modify, and distribute the software, with the stipulation that any derivative works give proper attribution to the original source. Moreover, the Apache License 2.0 includes explicit provisions for patent rights and contributes to the protection against copyright infringement, ensuring both users and contributors are safeguarded under a clear legal framework. This approach reflects our vision of making TimelinePTC a community-supported tool that benefits a wide array of users while maintaining the integrity and recognition of the creators' contributions.

## Use cases and applications

TimelinePTC, initially developed for the STEP clinic in New Haven, Connecticut, has recently been integrated into the program's growth to statewide coverage. This expansion has underscored the software's capability to scale effectively, meeting the growing demand for efficient data collection across a broader geographic area. Feedback from researchers and clinicians involved in data collection has been positive, including from those accustomed to the previous paper-based methods. Despite an initial adjustment period, users report feeling comfortable with TimelinePTC after just one session, highlighting its intuitive design and ease of use.

These frontline users report the most notable advantages of TimelinePTC being the speed with which data can be collected and the ability for real-time data review with participants. These features not only streamline the data collection process but also enhance the accuracy and completeness of the information gathered, as participants can directly verify and contribute to their data. This immediate feedback loop has been appreciated by those involved in the STEP program, emphasizing the tool's role in facilitating a more participant-engaged research approach.

From the data analysis perspective, the transition to TimelinePTC has brought about an improvement in efficiency and data quality. Previously, researchers faced the laborious task of manually extracting PTC data from stacks of paper forms, a process fraught with inefficiency and the potential for error. With TimelinePTC, this cumbersome step is eliminated, as the software automatically generates a comma-separated values (CSV) file that is ready for analysis. This automation has resulted in a substantial time saving, estimated at approximately 20–30 minutes per participant, drastically reducing the workload for the data analysis team and accelerating the pace at which research findings can be generated and applied. Also, when previously using the prior paper method, up to 10% of PTCs were excluded from post hoc analysis because they were internally inconsistent or there was insufficient data to fully reconstruct the full PTC [8]. The format and constraints of TimelinePTC make this outcome nearly impossible.

The high-quality, complete, and granular PTC data collected via TimelinePTC combine with its automatic, optimally formatted data output to enable powerful analytic products that were previously challenging to achieve. These data, whether analyzed individually or

concatenated for system-wide metrics, furnish actionable insights crucial for enhancing FEP services. System-wide network visualizations produced from these data allow for the detailed mapping and analysis of care pathways across the entire healthcare network. This visualization illuminates the complex interplay between all care nodes, identifying clustering, bottlenecks, and inefficiencies within the system [8 and Fig 2]. Additionally, the data collected with TimelinePTC facilitate highly granular marginal delay analysis, offering a nuanced understanding of how specific node subtypes contribute to delays in care [8 and S2 Table]. Such detailed analyses enable targeted interventions and optimizations within FEP services, directly addressing the factors that prolong DUP. Consequently, TimelinePTC stands as a pivotal tool in supporting FEP research and service delivery, providing a foundation for data-driven improvements and innovations in patient care pathways.

## Community nodes facilitating care:

**Self**–patient self-presented to the next node

**Education**—teacher or school counsellor facilitated

**Other**—community caregiver not otherwise included

## Clinical nodes providing care:

**ED**—Emergency Department

**Inpt**—Inpatient mental health admission

**Outpt**—Outpatient mental health provider

**IOP**—Intensive Outpatient Program [*A structured, non-residential treatment program for individuals with mental health or substance use disorders. It provides therapy and support multiple times a week for several hours each day, allowing participants to continue with their daily lives while receiving intensive treatment. IOPs are designed to offer a higher level of care than traditional outpatient therapy but are less intensive than inpatient programs, making them suitable for individuals who need significant support but do not require 24-hour supervision.*]

**Acute**—Acute evaluation [*An acute walk-in mental health service is a healthcare facility that provides immediate, on-demand mental health care for individuals experiencing severe or crisis-level mental health issues. These services offer rapid assessment, intervention, and support without the need for an appointment, helping to stabilize patients and direct them to appropriate ongoing care or treatment as needed. They serve as an alternative to emergency room visits for urgent mental health needs.*]

**PCP**—Primary Care Provider

**Mobile**—Mobile Evaluation [*A mental health assessment conducted by professionals who travel to the patient's location, such as their home, school, or community setting. This service is typically used in crisis situations to provide immediate evaluation and intervention, ensuring timely support and connecting individuals with appropriate resources and treatment plans without requiring them to visit a clinical facility. Mobile evaluations are often part of community mental health services aimed at increasing accessibility and responsiveness to urgent mental health needs.*]

## Clinical Nodes

## Community Nodes

**Fig 2. A directional graph of all PTCs collected in this study (from Mathis et al, 2022 [8]).** This is an example of the analytic output possible with robust collection of high-quality PTC data. Arrows depict the sequential progression individuals took from Onset to STEP enrollment. Clinical nodes are on the top, community nodes on the bottom. The thickness of the edge (line) between nodes reflects the frequency of traffic between them, and the size of each node reflects the cumulative number of interactions with the node type across all PTCs.

**OtherMH**–Other Mental Health Provider (e.g., Prison mental health, in-home psychiatric services, substance use disorder inpatient and outpatient)

**OtherMed**–Other Medical Provider, Outpatient non-psychiatric, non-PCP (e.g., neurologist), Inpatient medical.

## Discussion

TimelinePTC represents a significant advancement in the study of pathways to care (PTC). This web-based application, crafted with JavaScript and HTML, is designed to facilitate the process of collecting and visualizing PTC data in a dynamic, interactive, and collaborative manner. By transitioning from traditional paper-based methods to a digital platform, we have enhanced the accuracy, efficiency, and accessibility of PTC data collection.

A potential limitation to its current form is the local specificity of some of its data language. Developed within the specific context of the STEP program in Connecticut, some of TimelinePTC's language reflect aspects of the healthcare network of that area (e.g. the names of predetermined node types). However, its open-source nature and straightforward codebase invite adaptation and customization by other research teams across various healthcare contexts. The software's flexibility means that it can be readily modified to suit different settings or populations, broadening its applicability and potential impact.

TimelinePTC offers a powerful solution for enhancing data collection across larger, multi-site projects such as EPINET (Early Psychosis Intervention Network), a program dedicated to improving early intervention services for individuals experiencing first episode psychosis [16]. Its deployment is straightforward, as it operates on any device with a web browser, ensuring broad accessibility. The platform's thoughtful management of PHI and intuitive interface, requiring minimal training, make it user-friendly. Additionally, TimelinePTC provides immediate data output in CSV format, which seamlessly integrates with downstream analytics and industry-standard databases like REDCap. This capability can significantly improve data accuracy and efficiency in EPINET's early psychosis intervention efforts.

Looking forward, the implications of TimelinePTC's adoption extend far beyond immediate improvements in data collection efficiency. By making it easier for teams to adapt the tool to new contexts, TimelinePTC paves the way for a broader understanding of healthcare navigation across different diseases, demographic groups, and healthcare systems. This adaptability not only accelerates research in FEP but also opens doors to exploring patient pathways in other healthcare scenarios where understanding the transition from illness onset to clinical care is crucial (e.g. timelines of substance use or pathways to cancer care)

The long-term benefits of TimelinePTC's widespread use could support healthcare access research by providing more granular insights into the barriers and facilitators of care. Such insights are invaluable in designing interventions that are more targeted and effective at improving healthcare access and outcomes. Ultimately, TimelinePTC is an example of how technology can enhance research methodologies, offering a promising avenue for future explorations that could lead to substantial improvements in patient care and healthcare system efficiency.

## Conclusions

In summary, this web-based software offers a groundbreaking approach to collecting and visualizing pathways to care data. Through its interactive timeline, detailed event logging, and instant data export capabilities, the tool not only streamlines the research process but also enhances the quality and utility of the data collected, promising to advance the field of healthcare access research significantly.

## Supporting information

**S1 Table. Script for semi-structured PTC data collection with TimelinePTC.**
(DOCX)

**S2 Table. Descriptive statistics of node type encounter frequency and marginal-delay contribution (from Mathis et al, 2022 [8]).**
(DOCX)

## Acknowledgments

The authors would like to express their gratitude to Philip Markovich for his invaluable guidance and feedback during the development of this project.

## Author Contributions

**Conceptualization:** Walter S. Mathis, Maria Ferrara, John Cahill, Sneha Karmani, Vinod Srihari.

**Funding acquisition:** Vinod Srihari.

**Investigation:** Walter S. Mathis, Maria Ferrara, John Cahill, Sneha Karmani, Sümeyra N. Tayfur, Vinod Srihari.

**Methodology:** Walter S. Mathis, Maria Ferrara, Sneha Karmani, Sümeyra N. Tayfur, Vinod Srihari.

**Project administration:** Walter S. Mathis, Vinod Srihari.

**Resources:** Sneha Karmani, Vinod Srihari.

**Software:** Walter S. Mathis.

**Supervision:** Vinod Srihari.

**Validation:** Vinod Srihari.

**Visualization:** Walter S. Mathis.

**Writing – original draft:** Walter S. Mathis, Vinod Srihari.

**Writing – review & editing:** Walter S. Mathis, Sneha Karmani, Sümeyra N. Tayfur, Vinod Srihari.

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
