## [Decision Letter · Decision Letter 0]

24 May 2024

PONE-D-24-11289TimelinePTC: Development of a unified interface for pathways to care collection, visualization, and collaboration in first episode psychosisPLOS ONE

Dear Dr. Mathis,

Thank you for submitting your manuscript to PLOS ONE. After careful consideration, we feel that it has merit but does not fully meet PLOS ONE’s publication criteria as it currently stands. Therefore, we invite you to submit a revised version of the manuscript that addresses the points raised during the review process.

**please prepare a revision based on the comments**

We look forward to receiving your revised manuscript.

Kind regards,

Zhengmao Li

Academic Editor

PLOS ONE

Additional Editor Comments:

Please prepare a revision based on all comments

Reviewers' comments:

Reviewer's Responses to Questions

**Comments to the Author**

1. Is the manuscript technically sound, and do the data support the conclusions?

Reviewer #1: Yes

Reviewer #2: Yes

2. Has the statistical analysis been performed appropriately and rigorously? 

Reviewer #1: N/A

Reviewer #2: Yes

3. Have the authors made all data underlying the findings in their manuscript fully available?

Reviewer #1: Yes

Reviewer #2: Yes

4. Is the manuscript presented in an intelligible fashion and written in standard English?

Reviewer #1: Yes

Reviewer #2: Yes

5. Review Comments to the Author

Reviewer #1: TimelinePTC: Development of a unified interface for pathways to care collection, visualization, and collaboration in first episode psychosis

This is an interesting study that describes a tool that has been developed to easily visualize pathways to care in first episode psychosis patients. It highlights how an original tool has been modified to make it more efficient in data capture and data visualization.

Two issues that need resolving.

1. The paper has been written assuming that the reader knows a lot about specialized treatment early in psychosis programs. Therefore, the supplementary data include information that is not easy to understand for a non-specialist reader. The tool picks information from various sources that one has to know of to understand what it is doing. For example, it refers to the Structured Interview for Psychosis Risk Syndromes, intensive outpatient programs, mobile community teams etc. These are technical interventions that are not often routine care in psychosis care. There is need to explain these sources of information and what they do.

2. I do not know why the tables and figures are supplementary data. It would be better if they were included in the text. It makes reading and understanding easier.

Reviewer #2: Would like to see more thoughtful discussion on how piloting and implementation across Epinet Hubs could occur.

Nodes related to criminal justice (ieg Jail) could be added.

How are psychotropics along the pathway (ie not antipsychotics) considered?

How are the reasons for the encounter captured?

6. PLOS authors have the option to publish the peer review history of their article (what does this mean?). If published, this will include your full peer review and any attached files.

Reviewer #1: **Yes: **Dr. Emmanuel Kiiza Mwesiga

Reviewer #2: No

---

## [Author Response · Author response to Decision Letter 0]

14 Jun 2024

I have included response to reviewers in a separate file attached to this revision.

---

## [Decision Letter · Decision Letter 1]

5 Jul 2024

TimelinePTC: Development of a unified interface for pathways to care collection, visualization, and collaboration in first episode psychosis

PONE-D-24-11289R1

Dear Dr. Mathis,

We’re pleased to inform you that your manuscript has been judged scientifically suitable for publication and will be formally accepted for publication once it meets all outstanding technical requirements.

Kind regards,

Zhengmao Li

Academic Editor

PLOS ONE

Additional Editor Comments (optional):

Reviewers' comments:

Reviewer's Responses to Questions

**Comments to the Author**

1. If the authors have adequately addressed your comments raised in a previous round of review and you feel that this manuscript is now acceptable for publication, you may indicate that here to bypass the “Comments to the Author” section, enter your conflict of interest statement in the “Confidential to Editor” section, and submit your "Accept" recommendation.

Reviewer #1: All comments have been addressed

Reviewer #2: (No Response)

2. Is the manuscript technically sound, and do the data support the conclusions?

Reviewer #1: Yes

Reviewer #2: Yes

3. Has the statistical analysis been performed appropriately and rigorously? 

Reviewer #1: Yes

Reviewer #2: Yes

4. Have the authors made all data underlying the findings in their manuscript fully available?

Reviewer #1: Yes

Reviewer #2: Yes

5. Is the manuscript presented in an intelligible fashion and written in standard English?

Reviewer #1: Yes

Reviewer #2: Yes

6. Review Comments to the Author

Reviewer #1: (No Response)

Reviewer #2: Revision addressed questions.

7. PLOS authors have the option to publish the peer review history of their article (what does this mean?). If published, this will include your full peer review and any attached files.

Reviewer #1: **Yes: **Dr. Emmanuel Kiiza Mwesiga

Reviewer #2: No

---

## [Editor Report · Acceptance letter]

11 Jul 2024

PONE-D-24-11289R1 

PLOS ONE

Dear Dr. Mathis, 

I'm pleased to inform you that your manuscript has been deemed suitable for publication in PLOS ONE. Congratulations! Your manuscript is now being handed over to our production team.

Kind regards, 

on behalf of

Dr Zhengmao Li 

Academic Editor

PLOS ONE